# Trends and Prospects of Change in Wheat Self-Sufficiency in Egypt

Ahmed Abdalla [1,*], Till Stellmacher [1] and Mathias Becker [2]

1   Right Livelihood College (RLC), Center for Development Research (ZEF), University of Bonn,
    53113 Bonn, Germany
2   Institute for Crop Science and Resources Conservation (INRES), University of Bonn, 53113 Bonn, Germany
*   Correspondence: s5adabda@uni-bonn.de

**Abstract:** Egypt is the largest wheat importer in the world; however, it produces only half of the 20 million tons of wheat that it consumes annually. The population of Egypt is currently growing by 1.94% per year, and projections predict that the demand for wheat will be nearly doubled by 2050. Russia and Ukraine are major wheat exporters to Egypt and globally, shipping grains from ports in the Black Sea. The ongoing conflict aggravates the already precarious food security situation in Egypt and many other import-dependent countries in Africa and Asia by disrupting supplies and accelerating food price hikes. Wheat is a strategic commodity in Egypt. Its production is a question of political stability. Against this backdrop, the Egyptian government declared gaining wheat self-sufficiency as a strategic aim. This study provides an overview of the degree and trends of cultivated wheat area, yield, production, and wheat self-sufficiency in Egypt between 2000 and 2020, followed by a qualitative analysis determining external pressures and system-immanent drivers that had an impact on wheat self-sufficiency in the past two decades in view of predicting future pathways to achieve wheat self-sufficiency in a sustainable way. The study underlines some critical external pressures such as agricultural policies, (subsidized) production inputs, climate conditions, global wheat supply chains, and system-immanent drivers such as domestic wheat supply prices and yields influencing the area of wheat cultivation and its productivity. There is a significant need to implement more effective and long-term sustainable agricultural policies in order to make wheat production in Egypt (more) attractive and feasible for smallholders again.

**Keywords:** Egypt; wheat; food security; self-sufficiency; sustainable agriculture



## 1. Introduction

Increasing population growth and changing consumer demands require changes in agricultural production and systems. In 2050, the world's population is estimated to reach 9 billion people, requiring a 60% higher food production than in 2005–2007 [1]. The agricultural sector has to respond to this challenge. The required increase in primary food production is expected to come from productivity gains (higher yields and increased land-use intensities) rather than from area expansion [2].

Egypt is an agriculture-based country. However, 97% of Egyptian land is desert. Out of the total land area of about 100 million ha, only about 3.5 million ha is used for agricultural production, including the "new lands" reclaimed from the desert in the last decades [3]. In addition to the massive scarcity of arable land and water shortage, the population growth rate in Egypt is one of the highest worldwide. The population is currently growing at a rate of 1.94%, adding about 2 million people each year [4]. However, a favorable sub-tropical winter-rain climate combined with fertile soils in the Nile valley and water supply by the Nile River permits Egypt to produce a large variety of crops, ranging from coarse grain to fruits and vegetables and export-oriented crops such as citrus. Agriculture contributes approximately 14% to the total GDP and absorbs about 31% of the national

workforce. About 53% of the population lives in rural areas, with their livelihood directly or indirectly depending on agricultural activities [5]. Most farmers in Egypt are categorized as smallholders, with more than 80% of them owning less than 2 ha of cropland and 50% holding even less than 0.4 ha [5]. Smallholder agriculture contributes significantly to employment and food production, as well as land and water use.

Egypt's main crops in terms of cultivated area are wheat, rice, and a large diversity of vegetables and fruit trees (i.e., citrus and dates). Egypt also produces forage species for feeding ruminants (i.e., clover and alfalfa) and crops for export such as cotton, sugarcane, sunflower, and citrus [6,7]. The most important crops in the winter season are wheat, broad beans, onions, and alfalfa, whereas irrigated rice, maize, and cotton are the main crops in the summer season. However, the lack of arable land, the constantly increasing scarcity of water (which is likely to be worsened due to the construction of the Grand Ethiopian Renaissance Dam at the Blue Nile River in Ethiopia), and the high population growth rate, combined with land degradation, impacts of climate change, and rural poverty, are massively challenging the agricultural sector. Additionally, high production costs, poor irrigation infrastructure, and land fragmentations negatively impact production, particularly that of smallholders. Faced with these challenges, for many smallholders, especially the younger generation, agriculture is not attractive any longer. Many smallholders migrate to urban centers. All in all, Egypt's agricultural sector cannot meet the growing and changing demands, and the country heavily depends on the import of food, mainly wheat [8].

Wheat is historically the most important crop in Egypt. Egyptians derive one-third of their daily caloric intake and 45% of their protein intake from wheat-based food, mainly in the form of subsidized bread *baladi* [1]. Egypt consumes the equivalent of 18.5 million tons of wheat per year, while the per capita share reaches 196 kg per year [8]. The annual per capita consumption of wheat exceeds the global average by more than 100 kg. In 1977, hundreds of thousands of Egyptians were protesting after the president announced a cut of subsidies on bread and other essential commodities. About 77 people were killed and a large number were injured in these Egyptian "bread riots" [9]. In January 2011, during the Egyptian revolution, the repetitive cheers were "Bread, freedom and social justice", in which bread was the first need. The availability and quality of bread *baladi* were some of the main reasons for the Egyptian revolution in 2011 [10]. All in all, wheat is historically a strategic and highly political commodity in Egypt, and the subsidized bread prices have been unchanged for decades. The government stated the price of the bread *baladi* at a subsidized fixed price of EGP 0.05 per loaf (equivalent to USD 0.01 in January 2022), which is less than one-tenth of the actual production costs [11]. Over the years, the bread subsidy program has become a massive economic burden on the Egyptian state budget. For instance, in the fiscal year 2019/20 only, the government allocated EGP 89 billion (USD 5.69 billion) for food subsidies, and more than half of this went to subsidized bread. In the 2020/21 fiscal year, bread and food subsidies amounted to EGP 84.5 billion (USD 5.4 billion) [12]. The government directly subsidizes every level of the bread value chain from wheat procurement to flour milling to bakery production to maintain the final consumer price at 0.05 EGP per loaf [13]. The subsidies mainly aim to target and satisfy (poor and urban) consumers and not (poor and rural) producers. Despite the overall importance of wheat smallholders in Egypt, their situational and productional needs are largely neglected.

Egypt is the world's largest wheat-importing nation. Only less than half of the national consumption can be met by domestic production; the rest has to be covered by imports [14]. In 2018, about 12 million tons of wheat were imported [15]. In 2021, Russia and Ukraine contributed 85% of the total wheat imports to Egypt (60–66% depending on the years from the Russian Federation and 20–25% depending on years from Ukraine) [16], followed by Romania, France, and the United States [16]. The annual value of wheat imports was highest in 2014 at USD 5.9 billion and lowest in 2013 and 2016 at USD 2.6 billion [17]. Global food prices and specifically wheat prices have been rising since 2020 as the result of the global supply chain disruptions caused by the COVID-19 pandemic. For instance, wheat

prices averaged USD 280 per metric ton during the first half of 2021 and reached USD 317 per metric ton by November 2021. They were as high as around USD 500 per metric ton in February–April 2022 and will keep increasing because of the Russian-Ukrainian war [17]. Additionally, about 61% of the Egyptian population (63.5 million people) rely on subsidized *baladi* bread under a state-subsidized food card system [17]. The Egyptian government considered several measures aimed to fix the subsidized bread price at EGP 0.05 per and minimize its costs. Lastly, the government reduced the size of loaf from 110 g to 90 g, which is nearly 20%of its actual size, and excluded approximately 7.5 million people (about 10.6% of the whole population) from the subsidized food card system on the grounds that they can afford market prices [17]. The Egyptian government is seeking to apply new approaches to increase the quantities of flour extracted from domestic and imported wheat. For instance, the country is examining ways to obtain more flour from wheat grain by raising the extraction percentage for flour used for subsidized bread to 87.5 percent instead of 82 percent. Egypt also studied mixing barley with wheat to produce *baladi* bread [18]. However, barley production was insufficient as the crop is used extensively in beverages. More recently, another method was tested by mixing potatoes and wheat in a 50–50 mixture [18]. Against this backdrop, the vision of substituting very costly and increasingly insecure wheat imports by increasing domestic production was outlined by the Government of Egypt in 2014 in the "Egypt Sustainable Development Strategy Towards 2030". The strategy vision 2030 aims to increase domestic local production and sets goals to expand the cultivation of wheat. It includes vertical expansion to maximize productivity per unit and relies mainly on using new varieties of wheat with high productivity and implementing modern irrigation methods to minimize water consumption, and on the other hand, implementing horizontal expansion by cultivating new lands and using suitable varieties of wheat for the nature of these lands that can adapt to its climatic conditions, salinity, and drought [14]. The vision of the 2030 document emphasized the government's aspiration to attain 74% wheat self-sufficiency by 2017. However, the actual self-sufficiency ratio in 2017 was 43% [14]. Given the Russian-Ukrainian war and other international crises, the wheat import substitution strategy has gained further importance, and the global wheat market faces significant challenges for the future wheat supply, as the two countries account for about 30% of the world's wheat supply. Russia is the largest wheat exporter and Ukraine is among the largest producers.

The aim of this study is to estimate the degree and trends of wheat self-sufficiency, the area of production, and the productivity in Egypt between 2000 and 2020 and to determine the concerning external pressures and system-immanent drivers in order to achieve wheat self-sufficiency in a sustainable way. At first, a descriptive method was used to determine the levels and trends of the wheat self-sufficiency ratio, the cultivated area of wheat, and its production and productivity between 2000 and 2020 based on official and unpublished data collected from the Egyptian Ministry of Agriculture and Land Reclamation (MALR) and the Central Agency for Public Mobilization and Statistics (CAPMAS). Additionally, a content analysis of published data in the forms of reviewed literature, published national and international surveys and reports, regulatory documents, grey literature, and journal articles published over roughly the past two decades was conducted.

## 2. Material and Methods

This study mainly relies on quantitative and qualitative data based on various published and unpublished sources, mainly from the Egyptian Ministry of Agriculture and Land Reclamation (MALR) and the Central Agency for Public Mobilization and Statistics (CAPMAS), the United States Department of Agriculture (USDA), the Food and Agriculture Organization of the United Nations (FAO), and the World Bank. In addition, a content analysis of national and international surveys and reports, regulatory documents, grey literature, and journal articles published over roughly the past two decades was conducted. The data were analyzed using descriptive statistics. Frequency counts, percentages, and

means were used to describe the trends and prospects of wheat self-sufficiency, production, and yield.

## 3. Results

### 3.1. Wheat Demands and Consumption

Wheat demand and consumption depend on wheat price, consumers' income, prices of substitutes (e.g., barley, rice, and maize), and population growth [19]. The increasing population puts pressure on wheat demand and consumption, which directly contributes to various food vulnerability problems in Egypt. The population growth rate is faster than that of domestic wheat production in Egypt, which exerts pressure on wheat consumption. The Egyptian population in 2020 was estimated at 100 million and consumed about 20 million metric tons of wheat in the same year [20]. The Egyptian population is forecasted to reach 140 million by 2050, which will require about 30 million metric tons of wheat [19]. Furthermore, the annual wheat consumption per capita in Egypt is estimated at 196 kg annually, which is about 3 times more than the average global consumption per capita, which is estimated at 66.9 kg annually, and more than 3 times the average per capita consumption in developing countries, which estimated at 60.5 kg annually [8]. At the same time, one-third of the Egyptian population lives below the poverty line (with less than $1.97 per person a day), which has a significant role in increasing wheat demand [19]. The wheat gap between 2000 and 2020 increased annually by 6.5%, which is equivalent to 538 thousand tons annually [21] (Figure 1). The increasing wheat gap puts pressure on wheat imports, which causes food insecurity, especially when international wheat prices spike and the global supply chain is disturbed under different circumstances such as COVID-19 and the Russian-Ukrainian war. The average wheat gap in the period between 2000 and 2020 is estimated at 7715 thousand tons. The gap has ranged between a maximum gap in 2019 estimated at 12,936 thousand tons and a minimum gap estimated at 3564 thousand tons in 2001. The largest increase in the wheat gap was in 2010, with an estimated gap of 10,516 thousand tons from 6777 thousand tons in 2019, with a significant increase estimated at 3739 thousand tons. The lowest decrease in the wheat gap occurs in 2012, with a decrease in the wheat gap from 8782 thousand tons in 2011 to 6861 thousand tons in 2011, with an estimated gap of 1921 thousand tons.

### 3.2. Wheat Self-Sufficiency (2000–2020)

Despite the almost linear increase in the total wheat production in Egypt between 2000 and 2020, with an annual average of 1.64% and an annual growth average of productivity estimated at a rate of 1%, the wheat self-sufficiency rate shows an average declination from 2001 to 2020 of 2% annually (Figure 2). The maximum and minimum ratios of wheat self-sufficiency were 64% in 2001 and 40% in 2019. The largest decline in self-sufficiency occurred in 2010, being 27% lower than in 2009 by an estimated decline of 0.76 ton/ha. The highest increase in wheat self-sufficiency was 20% in 2011 with an estimated rise of 0.85 ton/ha from 5.69 ton/ha in 2010 to 6.54 ton/ha in 2011. The wheat self-sufficiency ratio is calculated as the percentage of domestic production from the total wheat supply total supply [12].

$$SSR = Production/(Production + imports - exports) \times 100$$

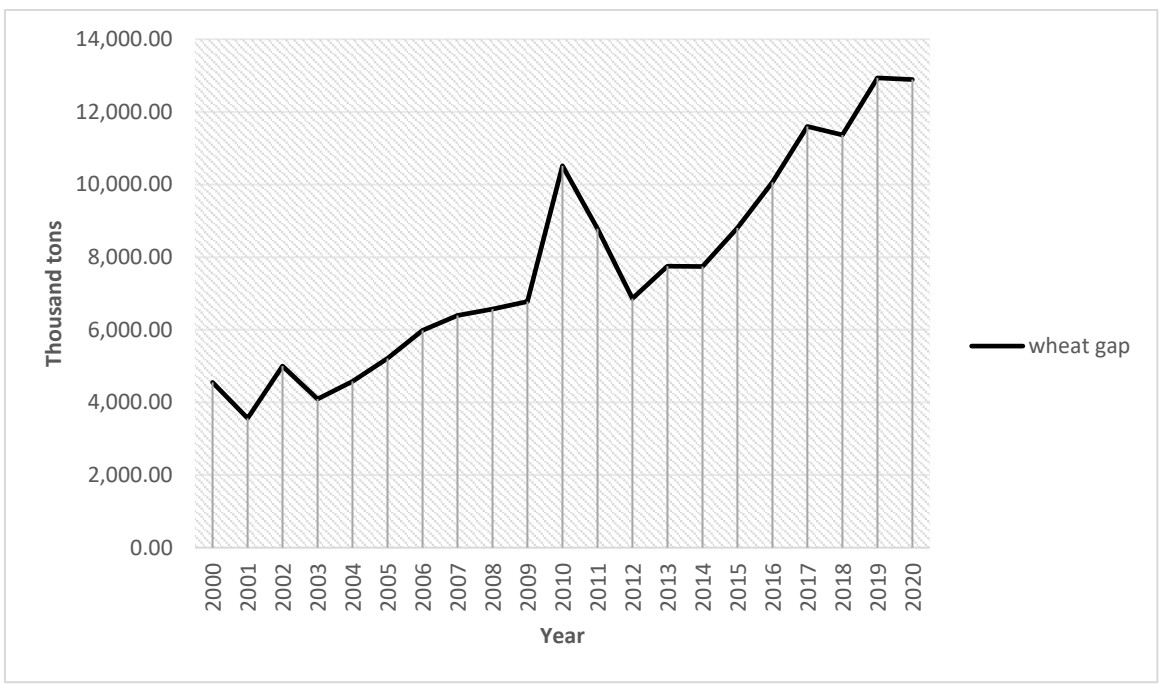

**Figure 1.** Trends of wheat gap in Egypt. Annual average percentage growth rate (2000–2020). Source: MALR & CAPMAS in Egypt and own elaboration [22,23].

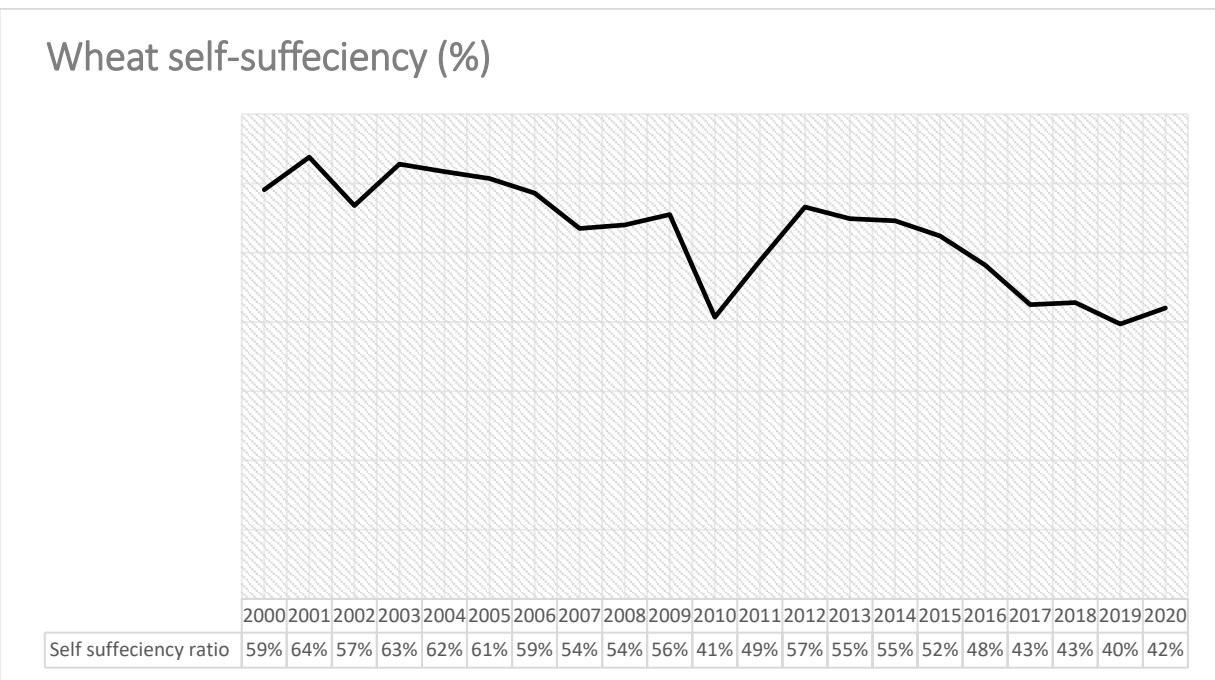

**Figure 2.** Trends of wheat self-sufficiency in Egypt. Annual average percentage growth rate (2000–2020). Source: MALR & CAPMAS in Egypt and own elaboration [22,23].

### 3.3. Wheat Production (2000–2020)

The total wheat production in Egypt between 2000 and 2020 has an average of 8.032 million tons with a 1.64% annual growth rate (Figure 3). The maximum total production was in 2015 at 9608 thousand tons, and the minimum production was in 2001 at 6255 thousand tons. The data show a decline in total production between 2016 and 2019 with an annual average decline rate of 3%. The total production decreased from 9607 thousand tons in 2015

to 8558 thousand tons in 2019. In contrast, the total wheat production rose rapidly from 2001 (6254 thousand tons) to 2006 (8274 thousand tons), with an average growth rate of 5%.

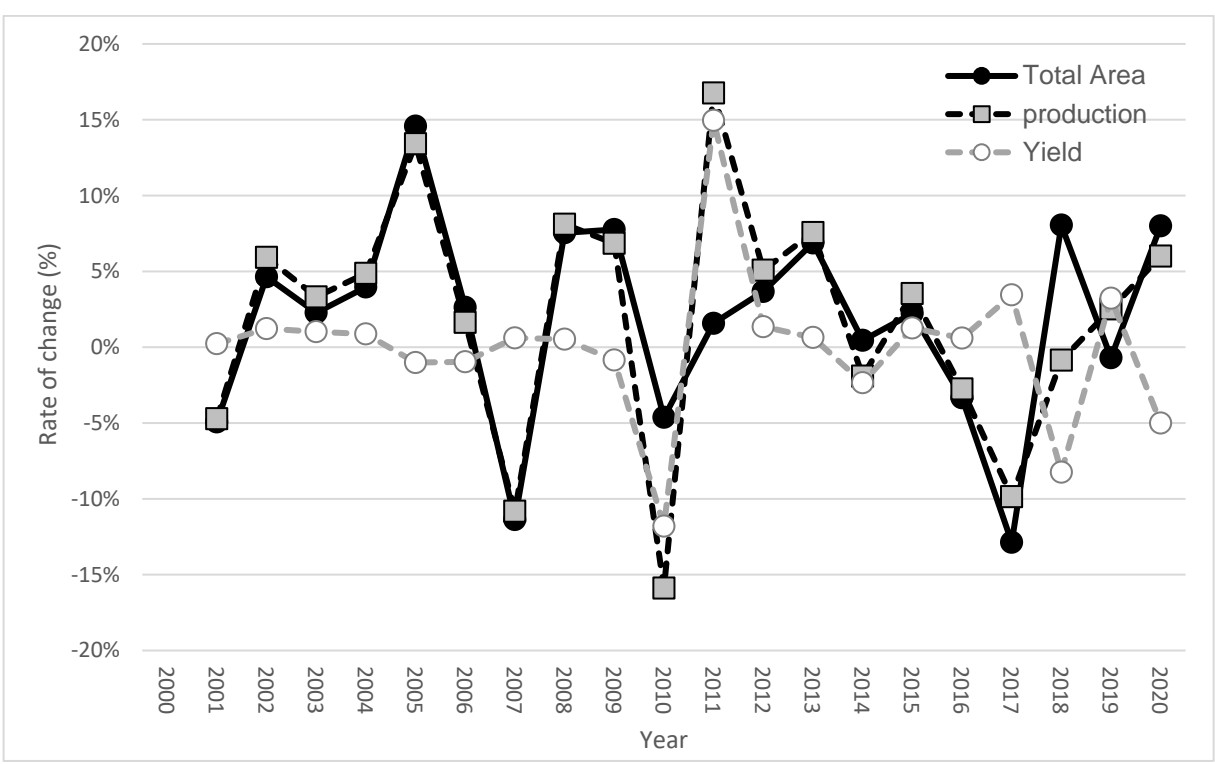

**Figure 3.** Trends in the total cultivated wheat area, wheat production, and yield of wheat in Egypt. Source: MALR & CAPMAS in Egypt and own elaboration. Annual average percentage growth rate (2000–2020) [22,23].

Meanwhile, the total production rose from 6254 thousand tons in 2001 to 8274 thousand tons in 2006, with an increase of 2020 thousand tons. The highest increases in total production were in 2005 (with 13% more than in 2014) and in 2011 (with 17% more than in 2010). In turn, the highest decreases were in 2007 (11% lower than in 2006) and in 2010 (16% lower than in 2009). Despite the increase in the total wheat area in 2018 by 8% more than in 2017, the yield declined by 6%, and consequently, the total production declined by 1%.

### 3.4. Wheat Cultivated Area (2000–2020)

The total cultivated wheat area in Egypt grew between 2000 and 2020 by 1.54% annually (Figure 3). The largest areas were cultivated in 2015 with 1456 thousand hectares and the smallest cultivated wheat area was in 2001 with 983 thousand hectares. The largest increases were in 2005 when the total wheat area grew by 21% from 1094 thousand hectares in 2004 to 1253 thousand hectares in 2005, followed by another increase in 2015 with an increase of 16% compared with 2014. In contrast, the total wheat area declined between 2015 and 2019, with a significant drop in 2017 (20% lower than in 2016).

### 3.5. Wheat Yield (2000–2020)

The average yield in Egypt between 2000 and 2020 per hectare is 6.47 tons/ha, with an annual growth rate of 0.2% (Figure 3). The maximum yield of 6.86 tons/ha was harvested in 2017 and the minimum yield of 5.69 ton/ha was harvested in 2010. The data show significant drops in yield/ha in 2010 and 2018. The significant drop in yield was in 2010 with a declination of 0.75 tons per hectare, followed by a yield spike in 2011 with an estimated increase of 0.85 tons per hectare. A notable declination occurred in the period

between 2017 and 2020, with a declination of 0.26 tons/ha (from 6.86 tons/ha in 2017 to 6.6 tons/ha in 2020).

### 3.6. Wheat Prices and Costs (2000–2020)

The price of domestic wheat supplied to the public silos in Egypt between 2000 and 2020 (EGP) increased annually by 71 EGP/ton (USD 4.5 in 2020), which is equivalent to 324 EGP/ha (USD 20.5 in 2020). In contrast, the total production costs increased by an average of 526 EGP/ha annually (USD 33.4 in 2022), while the maximum production costs were estimated at 27,710 EGP/ha in 2020 (USD 1755 in 2020). The lowest production costs were observed in 2005 with 7075 EGP/ha (USD 448). Between 2015 (10,905 EGP/ha (USD 1424) and 2020 (EGP 27,710 EGP/ha (USD 1755), the total production costs increased by 154%.

### 4. Discussion

The results reveal that the total wheat production in Egypt between 2000 and 2020 grew with an average growth rate of 1.64%, and the total wheat cultivated area grew in the same period with an average annual growth rate of 1.54%. In contrast, the wheat gap grew annually by an average growth rate of 6.5%, and wheat self-sufficiency grew by a 2% annual growth rate in the same period. The rapid population growth of 1.94% annually can be considered the main pressure on wheat self-sufficiency in Egypt. There are about 2 million more people every year and 392 thousand tons of additional wheat are needed annually, given an annual average consumption of 196 kg per capita. Despite the fact that the lowest total wheat production estimated at 6254 thousand tons and the smallest wheat cultivated area estimated at 983 thousand hectares between 2000 and 2020 was in 2001, the highest wheat self-sufficiency of 64% occurred in the same year. In 2019, Egypt had the lowest self-sufficiency rate of 40%, although with a total production of 8558 thousand tons on a total cultivated area of 1316 thousand hectares.

Figure 4 illustrates the total production of wheat in Egypt. The production started to rapidly rise from 2004 to 2005 as a result of the wheat self-sufficiency strategy implemented by the Minister of Agriculture and Land Reclamation at this time, Mr. Ahmed El-Lithy [15]. The strategy encouraged smallholders to grow more wheat by providing free machinery for land preparation and subsidized inputs such as seeds of high-yielding varieties [15]. The strategy also comprised strengthening agricultural extension. In combination, this resulted in a growth of total wheat area of 15% and the lowest total production costs estimated at EGP 7075 per hectare between 2000 and 2020. In 2005, the Egyptian government partially removed the subsidies for wheat growers, and new financial incentives for growing cash crops for export purposes were put in place. This reversed the so far positive trends in wheat production [15]. Wheat smallholders increasingly lacked access to machinery, agricultural services were reduced, and many production inputs were no longer subsidized. This led in 2007 to a significant drop of 11% in the wheat cultivated area and, consequently, to a significant drop in the total wheat production of 12%.

The world food crisis in 2007 led to a significant increase in international wheat prices. In Egypt, the food price spike led to demonstrations and violent conflicts that started in the industrial town of Mahalla, then spreading to other parts of the country [15]. This crisis refueled long-established arguments about the inefficiency of the government's food subsidy program at that time. The program relies in part on imported wheat and questions the government's ability to achieve wheat self-sufficiency [24]. In response to the crisis, the Egyptian government increased the national supply prices for wheat to encourage farmers to supply the increased amount of wheat to the national silos [8]. As direct result, the total production in Egypt increased in 2008 by 8%. In 2009, the farm-gate price of wheat declined again by 37%, which, together with an extreme heat wave in 2010, again led to a drop in the total production of 16% in 2010 and, in addition, led to significant drop in yield of 12% in the same year [25].

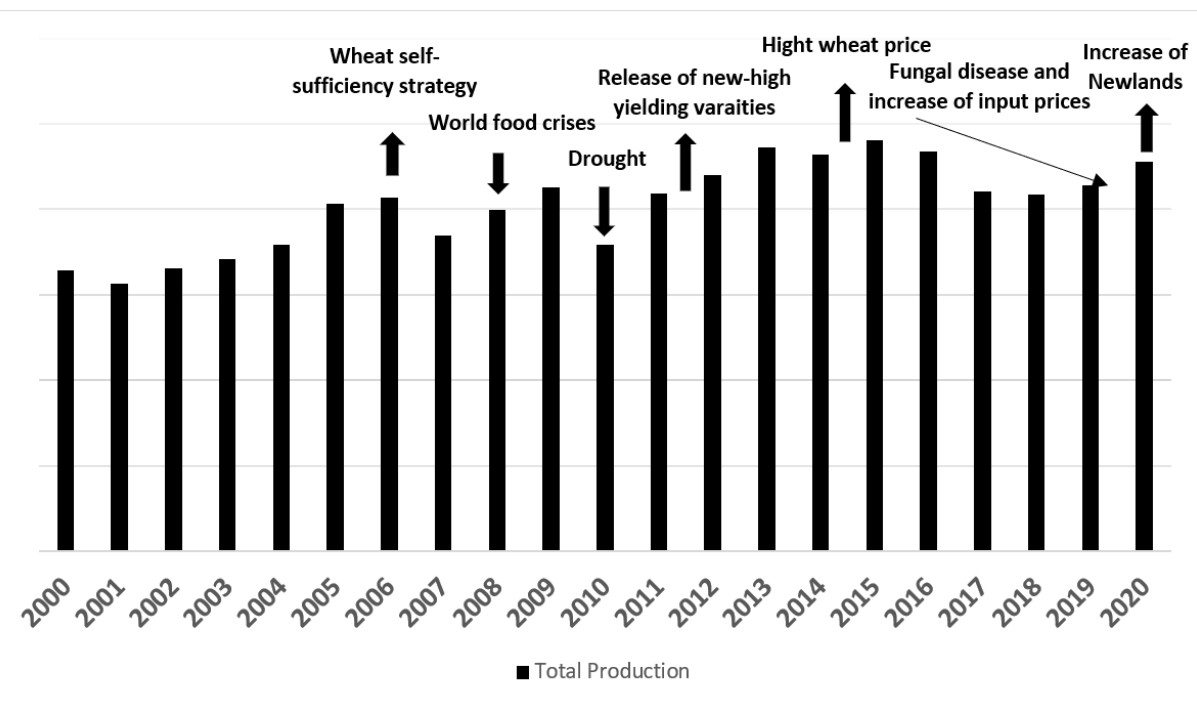

**Figure 4.** Trends and chronology of events that affected total wheat production in Egypt. (2000–2020). Source: MALR & CAPMAS in Egypt and own elaboration [22,23].

In 2010, the Egyptian government implemented the same policies as in 2007 and increased the wheat supply prices to encourage farmers to supply more wheat. This was subsequent to a decision of the Russian government to ban wheat exports to Egypt after millions of hectares withered in Russia due to drought and fires [15]. The resulting increase in global wheat prices led to a shortage of *baladi* in the Egyptian domestic market and, together with the lower quality, sparked new riots in Egypt in 2011 [10].

In 2016, the Egyptian government initiated an economic reform program that reduced parts of its energy subsidies, resulting in a significant increase in energy prices. Consequently, the production costs for wheat increased in 2017 by 34% compared with 2015 [21]. This led to a decline in the total cultivated wheat area of 13% and a 10% reduction in the total production. In 2018, despite the increase in the total cultivated area of 8% compared with 2017, the total production declined again by 1%, and the yield declined by 6%, largely due to wheat rust infections and the storm known in Egypt as "Dragon Storm" [25].

The time-series data analysis shows the main external pressures and system-immanent drivers that impacted wheat area, yield, and total wheat production in Egypt between 2000 and 2020. The rapid population growth together with the high poverty levels significantly increased bread demands. National agricultural policies tried to increase domestic wheat production by supporting wheat growers in the forms of subsidized inputs, suitable prices and extension services, and high-yield varieties, but policies were changed very often, sometimes from year to year, and rather half-heartedly implemented. International markets as well as climate conditions and diseases played another critical role.

One major factor influencing the total wheat cultivated area in Egypt is the time of declaration of the domestic wheat supply price (at the public silos gate). The Egyptian Ministry of Agriculture government often declares the domestic wheat price as late as the end of the growing season [15]. If the price is relatively high, this will encourage farmers to use more of their land for wheat production in the coming season; thus, the total cultivated area will increase. Therefore, producers' decisions are often determined in the short-term based on the previous year's price.

In fact, the largest share of the governmental budget for wheat subsidies is applied to bread production along the supply chain, from flour milling to bakeries [8]. In addition,

there is a substantial black market for wheat and flour, and in combination with corruption, part of the subsidized flour is sold on the black market, and other parts of wheat intended for human consumption end up as animal feed [26].

The prices that producers gain for their wheat produce compared with the production costs and the uncertainty due to the late declaration of the prices for the next season discourage many farmers from maintaining or even increasing their wheat production area or investing in higher productivity.

According to the Global Food Security Index, Egypt ranks 62nd out of 113 countries worldwide. Despite the Egyptian Sustainable Agriculture Strategy 2030 aiming at increasing productivity in the "old lands", the Egyptian government has been focusing in the last decades on the expansion of agricultural land into "new lands" through a "horizontal approach". For instance, between 2013 and 2022, the Egyptian government reclaimed 168.000 hectares in different land opening projects such as the "Four Million Acres Development Project", the "Northwest Coast Development Project", and the "West Minya Project" [27]. Yields in the "old lands" are mostly higher than those in the "new lands". Based on data from the MALR, the average wheat yields in the "old lands" is 6.68 t/ha compared with the maximum yield in the "new lands", which is 6.26 t/ha (see Figure 4). Moreover, developing "new lands" is protracted, very costly, and water-intensive, and the production process requires more fertilizers and other inputs compared with the "old lands" [28]. Wheat in the "new lands" is most often produced on larger farms by companies with contract workers, whereas wheat production in the "old lands" is almost entirely performed by smallholders [22,23,28]. Access to labor in the "new lands" is, however, difficult as they are often located in remote desert areas. Thus, the production costs are considerably higher while productivity is lower [29]. Consequently, producers in the "new lands" tend to grow cash crops for export instead of food crops such as wheat for domestic needs.

Egypt largely depends on importing wheat from Russia and Ukraine. Based on the official data from the Egyptian Ministry of Agriculture and Land Reclamation (MARL) and The Central Agency for Public Mobilization and Statistics (CAPMAS), in 2020, Egypt imported about 12.5 million tons of wheat. As a result of Russia's invasion of Ukraine in February 2022, Egypt lost its main sources of wheat. In reaction, in March 2022, the Egyptian government declared a new national strategy aimed to cope with this massive breakdown of its major wheat imports and the skyrocketing prices in the world market. The new strategy relies on increasing wheat self-sufficiency in Egypt by increasing the cultivated area and using more high-yield varieties on the one hand and finding new international wheat markets on the other [25]. However, the strategy does not mention what procedures should be taken and how this would practically help to strongly increase domestic wheat production in minimum time. It also does not say what incentives wheat-growing smallholders would receive to significantly increase their wheat production. Many of the policies in the wheat sector in Egypt are short-term and try to respond to immediate crises. What is basically lacking are overall concepts and the long-term implementation of these concepts that strengthen the national agri-food system's competitiveness, inclusiveness, and economic and ecological sustainability to ensure long-term food security in the country. Concepts have to provide incentives for smallholders, who are the backbone of food security in the country, to increase their wheat production and productivity and enable them to increase their income from wheat production. This would simultaneously reduce rural poverty and make agriculture more attractive to the next generation.

To be effective and sustainable, the policies have to be more long-term oriented and, particularly, based upon the concrete needs of the wheat-producing smallholders, mainly in the "old lands". Concrete support should include: (1) providing wheat-growing smallholders with tailored technical support and knowledge through agricultural extension, (2) providing wheat-growing smallholders with subsidized inputs on time, (3) declaring the national wheat prices before the growing season, and (4) determining higher domestic prices for domestic wheat supply. All these procedures combined would encourage and enable millions of experienced wheat-growing smallholders to increase their wheat-cultivated

area on their land and improve their productivity, which would directly lead to an increase in overall domestic wheat production. Additionally, given Egypt's very limited cropland and water resources, these measures would increase yields per unit in a "vertical approach", mainly in the "old lands", which can, however, go in parallel with a "horizontal approach" of reclaiming "new lands" for the production of other products, mainly for export.

## 5. Conclusions

With a population of about 104 million people [20], Egypt is the most populated country in the Arab world and the third most populated in Africa. Wheat is the most important food crop for the Egyptian diet and considered a highly strategic and political commodity in Egypt. Egyptians derive one-third of their daily caloric intake and 45% of their protein intake from wheat-based food, mainly in the form of the subsidized bread *baladi* [1]. Wheat is mainly produced by millions of smallholders along the River Nile and in the Nile delta. However, Egypt is the world's largest wheat-importing nation, with only less than half of the national consumption being met by domestic production. The social and political stability of Egypt depends on the availability of wheat and cheap *baladi*. The lack of access to affordable bread was one of the main reasons for the Egyptian revolution in 2011 in which the repetitive cheers were "Bread, freedom and social justice". In 2014, the Government of Egypt outlined the vision of substituting costly and insecure wheat imports by increasing domestic production. This strategy further gained pivotal importance since the significant increase in world market prices for wheat spurred by the Russian-Ukrainian war.

Against this backdrop, this paper aimed to determine the external pressures impacting wheat production and potentially wheat self-sufficiency in Egypt by analyzing the historical trends between 2000 and 2020. The findings of the paper show that although Egypt's wheat yields per unit of land are one of the highest globally, wheat self-sufficiency, however, declined annually by two percent on average from 2000 to 2020. Wheat production directly competes with cash crops such as broad beans and clover in old land and citrus, vegetables, and trees in new land, which are often more expensive to produce due to limited arable lands and scarce water resources. Most wheat is produced by smallholders in the "old lands". Wheat production in Egypt is characterized by high fluctuations in cultivated area, production, and yields. Policies are frequently changing, and wheat producers perceived a constant and massive decrease in support from the government and extension services between 2000 and 2020. Wheat-growing smallholders have no incentives to increase their wheat production given the shortage of subsidized inputs, inappropriate wheat prices, climate change, diseases, and increasing input prices. All in all, the attractiveness of wheat production, especially for the younger generation, declined between 2000 and 2020. The development of new agricultural land in the reclaimed "new lands" did not help to substantially increase domestic wheat production. Consequently, imports of wheat to Egypt were increasing between 2000 and 2020 instead of decreasing, and self-sufficiency is further predicted to decline in the next decade [30,31]. As Egypt's population rapidly grows and cultivable areas and available water are predicted to decrease, the country's reliance on wheat imports is likely to increase. One alternative would be to enhance the ability and incentives of millions of wheat-growing smallholders in the "old lands" to enhance their productivity. Based on this study this would need: (1) providing wheat-growing smallholders with technical support and knowledge through agricultural extension, (2) providing wheat-growing smallholders with subsidized inputs on time, (3) declaring the national wheat prices before the growing season, and (4) determining higher prices for domestic wheat supply. In short, to increase domestic wheat production, there is a significant need to implement an effective and long-term sustainable agricultural policy that makes wheat production (more) attractive and feasible for smallholders again.

**Funding:** This research was kindly funded through the Right Livelihood College (RLC) Campus Bonn by the German Academic Exchange Service (DAAD), grant number 91757798, and the Dr.

Hermann Eiselen Program of the Foundation fiat panis, grant number 91757798. The APC was funded by the University of Bonn.

**Acknowledgments:** We highly acknowledge the support of the German Academic Exchange Service (DAAD) and the Dr. Hermann Eiselen Programme of the Foundation fiat panis. We are thankful to our Egyptian partner SEKEM, who were awarded the Right Livelihood Award in 2003 for their excellent support during the field research.

**Conflicts of Interest:** The authors declare no conflict of interest.

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
