# Peer review of "Trends and Prospects of Change in Wheat Self-Sufficiency in Egypt"

_agriculture, doi:10.3390/agriculture13010007_

Round 1

Reviewer 1 Report

The article concerns a very important issue which is food self-sufficiency. This issue is important in the context of recent years, changing environment, war in Ukraine, rising production costs, inflation, etc. The article is interesting, but it needs to be supplemented to make it better perceived by foreign readers.

Technical notes:

- page 2: „EGP 0.05 per loaf (equivalent to US$ 0.01 in January 2022), which is less than one-tenth of the actual production costs (Wally et al., 2020).”  - What are per capita food expenditures and per capita incomes (in dollars) (both urban and rural)? (to show the price of this bread in relation to income or in a basket of food expenses)

- page 3 „Additionally, about 63.5 million people from the total population …” what is the percentage of the population?

- page 3 ” and excluded millions from the subsidised food card system…” what is the percentage of people who have previously received help?

- Figure 1- should include the amount of consumption and the amount of production - then it will be better to see where the gap in the wheat balance comes from (unless the gap results not only from production but also from insufficient imports - therefore a more accurate graph is needed)

- page 5, (3.2.) – „average of productivity”- how this productivity is measured?

- page 5 (3.2) – „wheat self-sufficiency rate” – You need to specify exactly self-sufficiency - is it the basic self-sufficiency ratio (SSR) -the relationship between domestic production and domestic use or relations between production and consumption

- page 6 (3.3) - please use the same denominations, preferably million tonnes because e.g. „in 2015 with 9.608 million tons…”  „9607 thousand tons in 2015”

- page 6- A figure should be inserted showing the crop area and yield per hectare in each year

- page 7 (3.6) - Dollars or another international currency, e.g. euro, should be used next to national units

- page 8 „In fact, the largest share of the governmental budget of the wheat subsidies is entitled to bread production along the supply chain, from flour milling to bakeries” (how much in elements of the supply chain?)

- The article cites only 11 items of scientific literature, i.e. scientific journals. There is a need to increase the amount of scientific literature and to show that the problem is not limited to a given country. But there are countries that have a similar problem with food self-sufficiency and you need to show how they deal with it. It is necessary to emphasize the effects of the war in Ukraine for various countries in terms of food security. There have been quite a few articles on this already.

Best regards

Author Response

Thank you for forwarding these helpful review reports. We are most grateful for
the time the you spent on providing suggestions on how to improve
our paper. In our revision, we have tried to address your suggestion as well as possible and specified in detail below

Point 1: This is interesting and timely but it lacks some contextual background, for example the importation of what from the Soviet block in the early 2000s and the role of FDI in wheat imports. Given the current crises in the Black Sea area I would encourage the authors to contextualise the problem of food insecurity in terms of the current global shortage of wheat supply, not wheat itself but supplies and supply chains.

The abstract needs some work to include more on methods  and results. An abstract needs to be clear, concise, comprehensive and correct.

So I recommend the above be addressed before sending out for review.

I would suggest shortening the abstract and not presenting statistics that are then presented in the Introduction section. The same with the Conclusions, which need to reflect the specifics of what you've found, rather than repeating the Introduction, or the material that should be in the Introduction.

Response 1: We thank you for this suggestion. We have amended the revised version of the manuscript accordingly and changed tha abstract as below:

Egypt is the largest wheat importer in the world, however, producing only half of the 20 million tons of wheat that it consumes annually. The population of Egypt is currently growing at 2% per year, and projections predict that the demand for wheat will double by 2050. Russia and Ukraine are major wheat exporters to Egypt and globally, shipping grains from ports in the Black Sea. The ongoing conflict aggravates the already precarious food security situation in Egypt and many other import-dependent countries in Africa and Asia by disrupting supplies and accelerating food price hikes. Wheat is a strategic commodity in Egypt. Its production is a question of political stability. Against this backdrop, the Egyptian government declared gaining wheat self-sufficiency as a strategic aim. This study provides an overview of the degree and trends of cultivated wheat area, yield, production, and wheat self-sufficiency in Egypt between 2000-2020, followed by a qualitative analysis determining external pressures and system-immanent drivers that impacted on wheat self-sufficiency in the past two decades in view of predicting future pathways to achieve wheat self-sufficiency in a sustainable way. The study underlines some critical external pressures such as agriculture policies, (subsidized) production inputs, climate conditions, global wheat supply chains, and system-immanent drivers such as domestic wheat supply prices, and yields influencing the area of wheat cultivation and its productivity. There is a significant need to implement more effective and long-term sustainable agriculture policies making wheat production in Egypt (again more) attractive and feasible for smallholders.

Point 2: page 2: „EGP 0.05 per loaf (equivalent to US$ 0.01 in January 2022), which is less than one-tenth of the actual production costs (Wally et al., 2020).”  - What are per capita food expenditures and per capita incomes (in dollars) (both urban and rural)? (to show the price of this bread in relation to income or in a basket of food expenses).

Response 2: This is a very interesting suggestion and we discussed it within our team. We feel that for the current paper we aim to focus to determine the external pressures and system immanent drivers impacting on the area of wheat cultivated and yield and productivity levels, yield, production, as well as the levels of self-sufficiency in Egypt. In our perspective food expenditures and per capita income do not appear to influence the above elements, including bread consumption due to the highly subsidized cheap price of Baladi bread. Additionally, these data have not been officially issued in the same year. We have included the suggestion into our current and ongoing work and once more, are thankful for having received this valuable idea from the reviewer.

Point 3: page 3 „Additionally, about 63.5 million people from the total population …” what is the percentage of the population?

Response 3: We thank you for this suggestion. We have amended our revised manuscript accordingly:

“Additionally, about 61% of the Egyptian population (63.5 million people) rely on subsidized baladi bread under a state subsidised food card system (The World Bank, 2022).”

Point 4: page 3 ” and excluded millions from the subsidised food card system…” what is the percentage of people who have previously received help?

Response 4: We thank you for this suggestion. We have amended the manuscript version and added the following text:

“excluded approximately 7.5 million people (about 10.6% of the whole population) from the subsidised food card system on the ground that they can afford market prices”

Point 5: Figure 1- should include the amount of consumption and the amount of production - then it will be better to see where the gap in the wheat balance comes from (unless the gap results not only from production but also from insufficient imports - therefore a more accurate graph is needed).

Response 5: Thank you for reviewer comment. According to our knowledge, our description of the methodological step/analysis/validation is appropriate in this case. The aim of this graph is to show only one trend (wheat gap) in the period between 2000 to 2020 which is equal to: (Wheat Gap = The total wheat consumption in million ton – The total domestic production in million ton) for each year. In our perspectives, this provides a clearer understanding, rather than showing a graph with two trends. In addition, the total domestic wheat production is also showing an annual increase, however the gap increases due to changes of other factors such as the population growth rate. We hope the reviewer and editor will follow our argument in this regard.

Point 6: page 5, (3.2.) – „average of productivity”- how this productivity is measured?

Response 6: Thank you for your question. All the quantitative data used in this publication are based on published and unpublished national official data sources, from the: 

  1. Federal Egyptian Ministry of Agriculture and Land Reclamation (MALR), and the
  2. Central Agency for Public Mobilization and Statistics 

In which productivity is calculated by dividing total production by the total cultivated area

Point 7: page 5 (3.2) – „wheat self-sufficiency rate” – You need to specify exactly self-sufficiency - is it the basic self-sufficiency ratio (SSR) -the relationship between domestic production and domestic use or relations between production and consumption

Response 7: We thank you for this suggestion. We have amended the revised version of the manuscript accordingly:

Wheat self-sufficiency ratio calculated as percentage of domestic production from the total wheat supply total supply

SSR = Production / (Production + imports – exports) × 100 (FAO, 2012)

https://www.fao.org/3/i2490e/i2490e00.htm

Point 8: page 6 (3.3) - please use the same denominations, preferably million tonnes because e.g. „in 2015 with 9.608 million tons…”  „9607 thousand tons in 2015”

Response 8: We thank you for this suggestion. We have amended the manuscript accordingly.

The maximum total production was in 2015 with 9,608 thousand tons and the minimum production was in 2001 with 6,255 thousand tons. The data show a decline in total production between 2016 and 2019 with an annual average decline rate of 3%. The total production decreased from 9,607 thousand tons in 2015 to 8,558 thousand tons in 2019. In contrast, the total wheat production rose rapidly from 2001 (6,254 thousand tons) to 2006 (8,274 thousand tons) with an average growth rate of 5%.

Point 9: page 6- A figure should be inserted showing the crop area and yield per hectare in each year.

Response 9: We thank you for this suggestion. We have amended our previous manuscript version and added the total wheat area and yield per hectare to the graph

Point 10: page 7 (3.6) - Dollars or another international currency, e.g. euro, should be used next to national units

Response 10: We thank you for this suggestion. We have amended our manuscript as follows:

“The price of domestic wheat supplied to the public silos in Egypt between 2000-2020 (EGP) increased annually by 71 EGP/ton (US$ 4.5 in 2020), which is equivalent to 324 EGP/ha (US$ 20.5 in 2020). In contrast, the total production costs increased by an average of 526 EGP /ha annually (US$ 33.4 in 2022) while the maximum production costs were estimated at 27,710 EGP/ha in 2020 (US$ 1755 in 2020). The lowest production costs were observed in 2005 with 7,075 EGP/ha (US$ 448). Between 2015 (10,905 EGP/ha (US$ 1424), and 2020 (EGP 27,710 EGP/ha (US$ 1755), the total production costs increased by 154%.”

Point 11: page 8 „In fact, the largest share of the governmental budget of the wheat subsidies is entitled to bread production along the supply chain, from flour milling to bakeries” (how much in elements of the supply chain?)

Response 11: Thank you for your suggestion to elaborate more on the aspect of the supply chain elements. The aim of this sentence is to highlight that wheat growers get no advantages from the bread subsidy system and all these subsidies support the players along the post-harvest value chain. As our research focus is mainly on the production side, specifically on smallholder farmers, we have decided not to expand the discussion on the value chain here. Instead, we have chosen to include that in anther paper in which we will include more detailed findings regarding the whole wheat value chain.

Point 12: The article cites only 11 items of scientific literature, i.e. scientific journals. There is a need to increase the amount of scientific literature and to show that the problem is not limited to a given country. But there are countries that have a similar problem with food self-sufficiency and you need to show how they deal with it. It is necessary to emphasize the effects of the war in Ukraine for various countries in terms of food security. There have been quite a few articles on this already.

Response 12: We are particularly grateful for your comment. This is a very interesting suggestion and we discussed it a lot within our team and added more items. Hoever, this paper is a part of a larger research project titled “Transition towards sustainable agriculture: A case study of wheat production systems in Egypt.” In this paper we aimed to determine the external pressures and system immanent drivers impacting on wheat self-sufficiency in Egypt. Wheat is a strategic commodity in Egypt. Its production is a question of political stability, It is characterized by subsidies, very high consumption per capita, and the fact that Egypt is the largest wheat importer in the world. Therefore, we prefer to focus our article on the Egyptian case and not the whole global wheat consumption and production problems. Last but not least, the quantitative data we used for the article was personally collected from official sources in Egypt. We appreciate your comment and we have included the suggestion into our current and ongoing work and once more, are thankful for having received this valuable idea from you.

Reviewer 2 Report

The article is very engaging, but I would like to see more adherence to the principles of writing specific parts of the article, e.g. I would suggest shortening the abstract and not presenting statistics that are then presented in the Introduction section. The same with the Conclusions, which need to reflect the specifics of what you've found, rather than repeating the Introduction, or the material that should be in the Introduction.

Some of the information is very repetitive. All repetitive material must be reviewed. The example:

Abstract: In 2021 80% of wheat imports in Egypt come from Russia and Ukraine.

Introduction: In 2021 Russia and Ukraine contributed with 85% of the total wheat imports to Egypt”.

Discussion: “Egypt largely depends on importing wheat from Russia and Ukraine.”; “In the same year 80% of wheat imports came from  Russia  and  Ukraine”.

Conclusions: “In 2021, 85% of wheat imports in Egypt came from Russia and Ukraine.

This message also needs to be concrete about from which country how much is imported.

Following your message: “The government directly subsidises every level of the bread value chain from wheat procurement to flour  milling to bakery production to maintain the  final consumer price at 0.05 EGP per  loaf (Kassim et al., 2018)”. The question is: what is the retail price of this bread without a subsidy?

Author Response

Thank you for forwarding these helpful review reports. We are most grateful for
the time you spent providing suggestions on how to improve our paper. In our revision, we have tried to address your suggestion as well as possible and specified in detail below.

Point 1: “The article is very engaging, but I would like to see more adherence to the principles of writing specific parts of the article, e.g. I would suggest shortening the abstract and not presenting statistics that are then presented in the Introduction section. The same with the Conclusions, which need to reflect the specifics of what you've found, rather than repeating the Introduction, or the material that should be in the Introduction”

Response 1: Thank you for the comment, we have excluded all the repetitions based on your suggestion and shortened the abstract as below:

Egypt is the largest wheat importer in the world, however, producing only half of the 20 million tons of wheat that it consumes annually. The population of Egypt is currently growing at 2% per year, and projections predict that the demand for wheat will double by 2050. Russia and Ukraine are major wheat exporters to Egypt and globally, shipping grains from ports in the Black Sea. The ongoing conflict aggravates the already precarious food security situation in Egypt and many other import-dependent countries in Africa and Asia by disrupting supplies and accelerating food price hikes. Wheat is a strategic commodity in Egypt. Its production is a question of political stability. Against this backdrop, the Egyptian government declared gaining wheat self-sufficiency as a strategic aim. This study provides an overview of the degree and trends of cultivated wheat area, yield, production, and wheat self-sufficiency in Egypt between 2000-2020, followed by a qualitative analysis determining external pressures and system-immanent drivers that impacted on wheat self-sufficiency in the past two decades in view of predicting future pathways to achieve wheat self-sufficiency in a sustainable way. The study underlines some critical external pressures such as agriculture policies, (subsidized) production inputs, climate conditions, global wheat supply chains, and system-immanent drivers such as domestic wheat supply prices, and yields influencing the area of wheat cultivation and its productivity. There is a significant need to implement more effective and long-term sustainable agriculture policies making wheat production in Egypt (again more) attractive and feasible for smallholders.

Point 2: Some of the information is very repetitive. All repetitive material must be reviewed. The example:

Abstract: “In 2021 80% of wheat imports in Egypt come from Russia and Ukraine.”

Introduction: In 2021 Russia and Ukraine contributed with 85% of the total wheat imports to Egypt”.

Discussion: “Egypt largely depends on importing wheat from Russia and Ukraine.”; “In the same year 80% of wheat imports came from  Russia  and  Ukraine”.

Conclusions: “In 2021, 85% of wheat imports in Egypt came from Russia and Ukraine.”

This message also needs to be concrete about from which country how much is imported.

Response 2: Thank you for the comment, we added the percentage for each country as below

Egypt imports approximately 85 percent of its wheat from the Russian Federation (60-66 percent depending on years) and from Ukraine (20-25 percent depending on years).

Point 3: Following your message: “The government directly subsidises every level of the bread value chain from wheat procurement to flour  milling to bakery production to maintain the  final consumer price at 0.05 EGP per  loaf (Kassim et al., 2018)”. The question is: what is the retail price of this bread without a subsidy?

Response 2: Thank you for your comment. The retail price in the private sector is 1 EGP. This is 20 times the price of the subsidised bread. However, the bread sold in the private sector is bigger in size, weight more, and better quality. Therefore, in our article, we cannot compare the prices of different breads. We hence prefer to focus on the bread called baladi that is most important for the daily diets of the majority of the Egyptian population, especially the urban and rural poor, and the political stability of the country.
